# Symbioses of Cyanobacteria in Marine Environments: Ecological Insights and Biotechnological Perspectives

**DOI:** 10.3390/md19040227

**Published:** 2021-04-16

**Authors:** Mirko Mutalipassi, Gennaro Riccio, Valerio Mazzella, Christian Galasso, Emanuele Somma, Antonia Chiarore, Donatella de Pascale, Valerio Zupo

**Affiliations:** 1Department of Marine Biotechnology, Stazione Zoologica Anton Dohrn, Villa Comunale, 80121 Naples, Italy; gennaro.riccio@szn.it (G.R.); christian.galasso@szn.it (C.G.); donatella.depascale@szn.it (D.d.P.); 2Department of Integrated Marine Ecology, Stazione Zoologica Anton Dohrn, Villa Comunale, 80121 Naples, Italy; valerio.mazzella@szn.it; 3Department of Life Sciences, University of Trieste, Via Licio Giorgieri, 34127 Trieste, Italy; emanuele.somma@szn.it; 4Department of Marine Biotechnology, Ischia Marine Centre, Stazione Zoologica Anton Dohrn, Punta San Pietro, 80077 Naples, Italy; valerio.zupo@szn.it; 5Department of Biology, University of Naples Federico II, Via Cinthia, 80126 Naples, Italy; antonia.chiarore@szn.it

**Keywords:** cyanobionts, diazotroph, secondary metabolites, animal interactions, prokaryotes, bioactive molecules, infochemicals

## Abstract

Cyanobacteria are a diversified phylum of nitrogen-fixing, photo-oxygenic bacteria able to colonize a wide array of environments. In addition to their fundamental role as diazotrophs, they produce a plethora of bioactive molecules, often as secondary metabolites, exhibiting various biological and ecological functions to be further investigated. Among all the identified species, cyanobacteria are capable to embrace symbiotic relationships in marine environments with organisms such as protozoans, macroalgae, seagrasses, and sponges, up to ascidians and other invertebrates. These symbioses have been demonstrated to dramatically change the cyanobacteria physiology, inducing the production of usually unexpressed bioactive molecules. Indeed, metabolic changes in cyanobacteria engaged in a symbiotic relationship are triggered by an exchange of infochemicals and activate silenced pathways. Drug discovery studies demonstrated that those molecules have interesting biotechnological perspectives. In this review, we explore the cyanobacterial symbioses in marine environments, considering them not only as diazotrophs but taking into consideration exchanges of infochemicals as well and emphasizing both the chemical ecology of relationship and the candidate biotechnological value for pharmaceutical and nutraceutical applications.

## 1. Introduction: Cyanobacteria and Their Symbiotic Associations

Cyanobacteria are a wide and diversified phylum of bacteria capable of photosynthesis. They are found in symbiosis with a remarkable variety of hosts, in a wide range of environments (Figure 1). Symbiotic relationships concern advantages and disadvantages for the organisms involved. Symbiosis, indeed, can be advantageous for only one of the involved organisms (commensalism, parasitism), or for both (mutualism) [1]. Symbiotic interactions are widespread and involve organisms among life domains, in both Eukaryota and Prokaryota (Archaea and Bacteria). Among prokaryotes, various species have been demonstrated to be associated with invertebrates such as sponges [2,3], corals [4,5,6,7], sea urchins [8], ascidians [9,10], and mollusks [11,12,13]. In addition, symbiotic relationships between bacteria and various microorganisms such as Retaria [14,15], Myzozoa [16], Ciliophora, and Bacillariophyceae [17] were investigated in the frame of the peculiar N_2_ fixing process performed by various associated prokaryotes. In fact, cyanobacteria are able to perform nitrogen fixation and, among all the symbiotic interactions they are able to establish, the nitrogenase products represent the major contribution to the partnership [18]. Nitrogen-fixing organisms are often called diazotrophs and their diazotroph-derived nitrogen (DDN) gives their hosts the advantage to populate nitrogen-limited environments [19,20]. Cyanobacterial symbionts (also named cyanobionts) are active producers of secondary metabolites and toxins [21], able to synthesize a large array of bioactive molecules, such as photoprotective and anti-grazing compounds [4,22]. In addition, cyanobionts have the advantage to be protected from environmental extreme conditions and from predation/grazing. In parallel, hosting organisms grant enough space to cyanobionts for growing at low competition levels. Several investigations demonstrated an influence of host organisms on the production of cyanobiont secondary metabolites, as in the case of the symbiotic interaction of *Nostoc* cyanobacteria with the terrestrial plant of *Gunnera* and *Blasia* genera [23]. Indeed, changes in the expression of secondary metabolites, as in the cases of the cyanobacterial nostopeptolide synthetase gene and the altered secretion of various nostopeptolide variants, were recorded in *Nostoc punctiforme* according to the presence of the host [24]. Changes in the metabolic profiles have probably a clear role in the formation of cyanobacterial motile filaments (hormogonia) and, most probably, they affect the infection process and the symbiotic relationship itself [24]. This suggests that cyanobacterial secondary metabolites may play a key role in host–cyanobacterium communications.

There are lines of evidence that cyanobionts produce novel compounds of interest to pharmaceutical research [25,26], exhibiting cytotoxic and antibacterial activities. Some of these molecules are produced by cyanobacteria only in a symbiotic relationship, as in the case of polyketide nosperin (Figure 2) [27].

Cyanobacteria are capable of establishing various types of symbiosis, with variable degrees of integration with the host, and probably symbiosis emerged independently with peculiar characteristics [28,29,30]. Symbionts are transferred to their hosts by a combination of vertical and horizontal transmission, with some strains passed down from ancestral lineage, while others are acquired by the surrounding environment [31]. However, cyanobacteria are less dependent on the host than other diazotrophs, such as rhizobia, due to the presence of specialized cells (i.e., heterocysts) and a cellular mechanism to reduce the oxygen concentration in the cytosol [32]. *Nostoc* species are heterocystic nitrogen-fixing cyanobacteria, producing motile filaments called hormogonia, and are considered the most common cyanobacteria in symbiotic associations [33,34]. The ability of diazotrophs cyanobacteria to fix nitrogen through various oxygen-sensitive enzymes, such as molybdenum nitrogenase (*nifH*), vanadium nitrogenase (*vnfH*), and iron-only nitrogenase (*anfH*), is a key point to fully understand the relationships between cyanobionts and their hosts [28]. 

Multicellular organisms coevolved with a plethora of symbiotic microorganisms. These associations have a crucial effect on the physiology of both [35] and, in some cases, the host-associated microbiota can be considered as a meta-organism forming an intimate functional entity [36]. This means that there are coevolutive factors that led to the evolution of signals, receptors, and infochemicals among the organisms involved in symbiosis. Host–symbionts communication, based on this complex set of dose-dependent [37] and evolutionarily evolved [38] infochemicals, influences many physiological aspects of symbiosis; some examples are the microbiota composition, defensive mechanisms, development, morphology, and behavior (Figure 3) [39]. The main interactions occurring between cyanobacteria and host organisms are summarized in Table 1.

## 2. Protists

Photosynthetic eukaryotes are the product of an endosymbiotic event in the Proterozoic oceans, more than 1.5 billion years ago [86,87]. For this reason, all eukaryotic phytoplankton can be considered an evolutive product of symbiotic interactions [87] and the chloroplast, as the remnant of an early symbiosis with cyanobacteria [86]. Nowadays, the associations among these unicellular microorganisms range from simple interactions among cells in close physical proximity, often termed “phycosphere” [88], to real ecto- and endosymbiosis. The study of these associations is often neglected, partially because symbiotic microalgae and their partners show an enigmatic life cycle. In most of these partnerships, it is unclear whether the relationships among partners are obligate or facultative [89]. The symbiotic associations between cyanobacteria and planktonic unicellular eukaryotes, both unicellular and filamentous, are widespread, in particular in low-nutrient basins [89]. It is assumed that cyanobacteria provide organic carbon through photosynthesis, taking advantage of the special environmental conditions offered by the host. In contrast, some single-celled algae are in symbiotic association with diazotrophic cyanobacteria, providing nitrogen-derived metabolites through N_2_ fixation [90]. This exchange is important for nitrogen acquisition in those environments where it represents a limiting factor, both in terrestrial and in aquatic systems, as well as in open oceans [91]. In fact, in marine environments, cyanobacteria are associated with single-celled organisms such as diatoms, dinoflagellates, radiolarians, and tintinnids [52,92]. The exchange of nitrogen between microalgae and cyanobacterial symbionts, although important, is probably flaked by other benefits such as the production of metabolites, vitamins, and trace elements [49,93]. In fact, available genomic sequences indicate bacteria, archaea, and marine cyanobacteria as potential producers of vitamins [94], molecules fundamental in many symbiotic relationships. Moreover, about half of the investigated microalgae have to face a lack of cobalamin, and other species require thiamine, B_12,_ and/or biotin [95,96]; these needs may be satisfied, in many cases, by the presence of cyanobionts [97].

The first case described of marine planktonic symbiosis was represented by the diatom diazotrophic associations (DDAs) among diatoms and filamentous cyanobacteria provided of heterocysts [98]. Although this kind of interaction is the most studied, little is known about the functional relationships of the symbiosis. Recent studies are mainly focused on the symbiotic relationships between the diazotroph cyanobacteria *Richelia intracellularis* and *Calothrix rhizosoleniae* with several diatom partners, especially belonging to the genera *Rhizosolenia*, *Hemiaulus*, *Guinardia*, and *Chaetoceros* [18,40]. The location of the symbionts varies from externally attached to partially or fully integrated into the host [41]. Indeed, it has been demonstrated through molecular approaches that morphology, cellular location, and abundances of symbiotic cyanobacteria differ depending on the host and that the symbiotic dependency and the location of the cyanobionts *R. intracellularis* and *C. rhizosoleniae* seems to be linked to their genomic evolution [99]. In this regard, it was demonstrated a clear relationship between the symbiosis of diatom–cyanobacteria symbiosis and the variation of season and latitude suggesting that diatoms belonging to the genus *Rhizosolenia* and *Hemiaulus* need a symbiont for high growth rates [40]. The reliance of the host seems closely related to the physical integration of symbionts: endosymbiotic relationships are mainly obligatory, while ecto-symbiosis associations tend to be more facultative and/or temporary [89]. Another interesting cyanobacteria–diatoms symbiosis involves the chain-forming diatom *Climacodium frauenfeldianum*, common in oligotrophic tropical and subtropical waters [100]. In this case, diatoms establish symbiotic relationships with a coccoid unicellular diazotroph cyanobacterial partner that is similar to *Crocosphaera watsonii* in morphology, pigmentation, and nucleotide sequence (16S rRNA and *nifH* gene) [41]. In addition, it has been demonstrated that nitrogen, fixed by cyanobionts is transferred to diatom cells [90]. Occasionally, *C. watsonii* has been reported as symbiotic diazotroph in other marine chain-forming planktonic diatoms, such as those belonging to the genera *Streptotheca* and *Neostrepthotheca* [42]. One of the most peculiar symbiosis is represented by the three-part partnership between the unicellular cyanobacterium *Synechococcus* sp., *Leptocylindrus mediterraneus*, a chain-forming centric diatom, and *Solenicola setigera*, an aplastidic colonial protozoa [43,44]. This peculiar association is cosmopolitan and occurs primarily in the open ocean and the eastern Arabian Sea; nevertheless, it remained poorly studied and exclusively investigated by means of microscopy techniques. Electron microscopy observations (SEM) reveal that in presence of *S. setigera*, the diatom can be apochlorotic (it lacks chloroplasts), thus offering refuge to the aplastidic protozoan, benefiting, and nourishing from the exudates it produces. It is assumed that the cyanobacterial partner, *Synechoccus* sp., supports the protozoan by supplying reduced nitrogen. It is also speculated that the absence of the cellular content of *L. mediterraneus* can be due to parasitism by *S. setigera* [44]. Recent studies reported a novel symbiotic relationship between an uncultivated N_2_-fixing cyanobacterium and a haptophyte host [45,46,47,48,49]. The host is represented by at least three distinctly different strains in the *Braarudosphaera bigelowii* group, a calcareous haptophyte belonging to the class of Prymnesiophyceae [101,102,103]. The cyanobiont, first identified in the subtropical Pacific Ocean through the analysis of *nifH* gene sequence, is UCYN-A or “*Candidatus Atelocyanobacterium Thalassa*,” formerly known as Group A. For many years, the lifestyle and ecology of this cyanobiont remained unknown, because cannot be visualized through fluorescence microscopy. Furthermore, the daytime maximum *nifH* gene expression of UCYN-A opposite with respect to unicellular diazotroph organisms [104,105]. The entire genome of the UCYN-A cells was sequenced, leading to the discovery of the symbiosis: the genome is unusually small (1.44 Mbp) and revealed unusual gene deletions, suggesting a symbiotic life history. Indeed, the genome completely lacks some metabolic pathways, oxygen-evolving photosystem II (PSII), RuBisCo for CO_2_ fixation, and tricarboxylic acid (TCA), revealing that the cyanobiont could be a host-dependent symbiont [47,48].

Symbiotic relationships include interactions between cyanobacteria and nonphototrophic protists. Heterotrophic protists include nonphotosynthetic, photosynthetic and mixotrophic dinoflagellates, radiolarians, tintinnidis, silicoflagellates, and thecate amoebae [51,52,92,106,107]. In dinoflagellates, cyanobionts were observed using transmission electron microscopy with evidence of no visible cell degradation, the presence of storage bodies and cyanophycin granules, nitrogenase, and phycoerythrin (confirmed by antisera localization), confirming that these cyanobionts are living and active and not simple grazed prey [52,108,109]. In addition, these cyanobionts are often observed with coexisting bacteria, suggesting a potential tripartite symbiotic interaction [52,109]. A cyanobiont surrounding the outer sheath was observed in rare cases, suggesting an adaptation to avoid cell degradation in symbiosis [52]. Despite the presence of N_2_ fixing cyanobacteria, molecular analyses demonstrated the presence of a vast majority of phototrophic cyanobionts with high similarity to *Synechococcus* spp. and *Prochlorococcus* spp. [50,51]. The complex assemblage of cyanobacteria and N_2_ fixing proteobacteria suggests a puzzling chemical and physiological relationship among the components of symbiosis in dinoflagellates, with an exchange of biochemical substrates and infochemicals, and the consequent coevolution of mechanisms of recognition and intracellular management of the symbionts. In tintinnid, ciliates able to perform kleptoplastidy, epifluorescent observations of *Codonella* species demonstrated the presence of cyanobionts, with high similarities with *Synechococcus*, in the oral grove of the lorica and, in addition, the presence of two bacterial morphotypes [52]. In radiolarians (Spongodiscidae *Dictyocoryne truncatum*), the presence of cyanobionts has been demonstrated, initially identified as bacteria or brown algae [110,111]. In addition, several non-N_2_-fixing cyanobionts have been identified using autofluorescence, 16s rRna sequence, and cell morphology, resembling *Synecococcus* species [51,52]. In agreement with associations observed in dinoflagellates, mixed populations of cyanobacteria and bacteria are common in radiolarian species, although their inter-relationship is still unknown. 

## 3. Macroalgae and Seagrasses

Mutual symbioses between plants and cyanobacteria have been demonstrated in macroalgae and seagrasses, as is the case of *Acaryochloris marina* and *Lynbya* sp., in which cyanobacteria contribute to the epiphytic microbiome of the red macroalgae *Ahnfeltiopsis flabelliformis* [53] and *Acanthophora spicifera* [54], respectively. Epiphytic relationships have been demonstrated as well with green and brown algae [112]. 

In *Codium decorticatum*, endosymbionts cyanobacteria belonging to genera *Calothrix*, *Anabaena*, and *Phormidium*, have been shown to fix nitrogen for their hosts [55,56].

Cyanobacteria are also common as seagrass epiphytes, for example, on *Thalassia testudinum*, where organic carbon is produced by cyanobacteria and other epiphyte symbiotic organisms rather than the plant itself [57,58]. In many cases, the presence of phosphates stimulates the cyanobionts growth on seagrasses and other epiphytes [113,114]. In oligotrophic environments, nitrogen-fixing cyanobacteria are advantaged against other seagrass algal epiphytes [115], and these cyanobacteria may contribute to the productivity of seagrass beds [116]. In addition, a certain level of host specificity can be determined in many plant–cyanobacteria symbioses [59], for example, among heterocystous cyanobacteria such as *Calothrix* and *Anabaena*, and the seagrass *Cymodocea rotundata*. A few cyanolichens live in marine littoral waters [92], and they play a role in the trophism of Antarctic environments, where nitrogen inputs from atmospheric deposition are low [117,118,119].

## 4. Sponges

Marine sponges are among the oldest sessile metazoans, known to host dense microbial communities that can account for up to 40–50% of the total body weight [31]. These microbial communities are highly species-specific, and characterized by the presence of several bacterial phyla; cyanobacteria constitute one of the most important groups [120,121,122]. Sponges with cyanobionts symbionts can be classified as phototrophs when they are strictly depending on symbionts for nutrition or mixotrophs when they feed also by filter feeding [92]. These “cyanosponges” are morphologically divided into two categories—the phototrophs present a flattened shape, while the mixotrophs have a smaller surface area to volume ratio [29]. Cyanobacteria are located in three main compartments in sponges: free in the mesohyl, singly or as pairs in closed-cell vacuoles, or aggregated in large specialized “cyanocytes” [123]. Their abundance decreases away from the ectosome, while it is null in the endosome of the sponge host [124]. Cyanobacteria belonging to the genera *Aphanocapsa, Synechocystis, Oscillatoria*, and *Phormidium* are usually found in association with sponges and most species are located extracellularly, while others have been found as intracellular symbionts benefiting sponges through fixation of atmospheric nitrogen [92]. Indeed, some cyanobacteria located intracellularly within sponges showed to own nitrogenase activity [124]. Most of the sponges containing cyanobionts, however, are considered to be net primary producers [125]. Cyanobacteria in sponges can be transmitted vertically (directly to the progeny) or horizontally (acquired from the surrounding environment), depending on the sponge species [29]. For instance, the sponge *Chondrilla australiensis* has been discovered to host cyanobacteria in its developing eggs [126]. Caroppo et al., instead, isolated the cyanobacterium *Halomicronema metazoicum* from the Mediterranean sponge *Petrosia ficiformis*, which has been later found as a free organism and isolated from leaves of the seagrass *Posidonia oceanica* [119,127], highlighting that horizontal transmission of photosymbionts can occur in other sponge species [128]. Cyanobacteria associated with sponges are polyphyletic and mostly belonging to *Synechoccoccus* and *Prochlorococcus* genera [129]. *Synechococcus spongiarum* is one of the most abundant symbionts found in association with sponges worldwide [130,131]. In some cases, however, the relationship between symbionts and host sponges can be controversial. Some *Synechococcus* strains seem to be mostly “commensals”, whereas symbionts from the genus *Oscillatoria* are involved in mutualistic associations with sponges [3,132]. 

In the past, many researchers performed manipulative experiments to demonstrate the importance of cyanobacteria associations for the metabolism of the host [3,128,133]. A case study from Arillo et al. performed on Mediterranean sponges revealed that *Chondrilla nucula*, after six months in the absence of light, displayed metabolic collapse and thiol depletion [63]. This highlights that symbionts are involved in controlling the redox potential of the host cells transferring fixed carbon in the form of glycerol 3-phosphate and other organic phosphates. Instead, *Petrosia ficiformis*, which is known to live in association with the cyanobacterium *Aphanocapsa feldmannii* [62], showed the capability to perform heterotrophic metabolism when transplanted in dark conditions [63]. In some tropical environments, the carbon produced by cyanobionts can supply more than 50% of the energy requirements of the sponge holobiont [122]. Cyanobacteria, moreover, can contribute to the sponge pigmentation and production of secondary metabolites (e.g*.*, defensive substances) [134], as in the case of the marine sponge *Dysidea herbacea* [64]. Thus, symbiotic associations could result in the production of useful compounds with biotechnological potential [134,135]. Meta-analysis studies on sponge–cyanobacterial associations revealed that several sponge classes could host cyanobacteria, although most of the knowledge in this field remains still unknown, and mostly hidden in metagenomics studies [136]. Sponge-associated cyanobacteria hide a reservoir of compounds with biological activity, highlighting an extraordinary metabolic potential to produce bioactive molecules for further biotechnological purposes [137].

## 5. Cnidarians

It is widely accepted that reef environments rely on both internal cycling and nutrient conservation to face the lack of nutrients in tropical oligotrophic water [138]. A positive ratio in the nitrogen export/input between coral reefs and surrounding oceans has been observed [139,140]. Tropical Scleractinia are able to obtain nitrogen due to various mechanisms that include the endosymbiont *Symbiodinium* [141], the uptake of urea and ammonium from the surrounding environment [142], predation and ingestion of nitrogen-rich particles [143,144,145,146], or diazotrophs itself through heterotrophic feeding [147] and nitrogen fixation by symbiotic diazotrophic communities [4,7,68,69,73,148]. In addition to nitrogen fixation, coral-associated microbiota performs various metabolic functions in carbon, phosphorus, sulfur, and nitrogen cycles [74,149,150,151]; moreover, it plays a protective role for the holobiont [152,153,154], possessing inhibitory activities toward known coral pathogens [155]. These complex microbial communities that populate coral surface mucopolysaccharide layers show a vertical stratification of population resembling the structure of microbial mats, with a not-dissimilar flux of organic and inorganic nutrients [156]. It is reasonable to believe that microbiota from all the compartments, such as tissues and mucus, can contribute to the host fitness and interact with coral in different ways, ranging from the direct transfer of fixed nitrogen in excess to the ingestion and digestion of prokaryotes [20].

Diazotrophs, and in particular cyanobionts, are capable of nitrogen fixation and they can use glycerol, produced by zooxanthellae, for their metabolic needs [4,73]. The relationship between corals and cyanobacteria is yet to be fully explored and understood but some lines of evidence regarding *Acropora millepora* [69,70] suggest coevolution between corals and associate diazotrophs (cyanobionts). This relationship appears to be highly species-specific. In hermatypic corals, a three-species symbiosis can be observed, with diazotrophs in direct relation with *Symbionidium* symbiont. In *Acropora hyacinthus* and *Acropora cytherea*, cyanobacteria-like cells, characterized by irregular layered thylakoid membranes and with a remarkable similarity to the ones described by previous authors [4], were identified in strict association with *Symbiodinium*, within a single host cell, especially in gastrodermal tissues [67]. The high density of these cells closely associated with *Symbiodinium* suggests that the latter is the main user of the nitrogen compounds produced by the cyanobacterium-like cells. The presence of these cyanobacterium-like cells is more widespread than assumed in the past and this symbiosis was found in many geographic areas, for example, in the Caribbean region and the Great Barrier Reef [67].

Microbial communities inhabiting the coral surface can greatly vary due to environmental conditions [147,157,158]. Diazotroph-derived nitrogen assimilation by corals varies on the basis of the autotrophic/heterotrophic status of the coral holobiont and with phosphate availability in seawater. Consequently, microbial communities increase when corals rely more on heterotrophy or when they live in phosphate-rich waters [147]. This suggests that diazotrophs can be acquired and their population managed according to the needs of corals [159]. This view was confirmed by the identification of a first group of organisms that form a species–specific, temporarily, and spatially stable core microbiota and a second group of prokaryotes that changes according to environmental conditions and in accordance with the host species and physiology state [160]. Experimental lines of evidence, using N_2_-labelled bacteria, demonstrated that diazotrophs are transferred horizontally and very early in the life cycle, and it is possible to identify *nifH* sequences, in larvae and in one-week-old juveniles [70], and in adult individuals [69] of the stony coral *Acropora millepora.* About coral tissues, the distribution of microbiota, and cyanobacteria as well, is not the same in all the tissue districts. Species that live in the mucus resemble the species variety and abundance that can be found in the surrounding water. On the contrary, the microbiota of internal tissues including also calcium carbonate skeletons is made, at least partially, of species that cannot be easily found free in the environment [68,69]. This plasticity might as well characterize cyanobacteria hosted in cnidarians, although such multiple relationships are still scarcely investigated.

*Synechococcus* and *Prochlorococcus* cyanobacteria have been identified in association with *Montastraea cavernosa* [4], through molecular approaches and genes belonging to filamentous cyanobacteria [6]. Filamentous and unicellular diazotrophic cyanobacteria belonging to the orders Chroococcales, Nostocales, Oscillatoriales, and Proclorales were found, using pyrosequencing approach, as associated organisms to the shallow water coral *Porites astreoides* [6] and *Isopora palifera* [71]. On the contrary, in *Montipora flabellate, Montipora capitate* [7], *Acropora millepora* [69,70], *Acropora muricate*, and *Pocillopora damicornis* [69], cyanobacteria are present in various tissues and in the skeleton, but their contribution in terms of nitrogen fixation is minimal [5]. In *Montastraea cavernosa*, *Montastraea franksi*, and in species of the genus *Diploria* and *Porites*, cyanobacterial sequences belonging to various genera (e.g., *Anabaena*, *Synechoccus*, *Spirulina*, *Trichodesmium*, *Lyngbya*, and *Phormidium*) have been found in coral tissues by PCR amplification [4,73,74,75,161]. In *Montastraea cavernosa*, the orange fluorescence protein, peaking at 580 nm, was attributed to phycoerythrin, a cyanobacterial photopigment produced by a cyanobacterium living in the host epithelial cells [4]. The different colors, especially of fluorescent proteins in corals, suggest specific biological functions for these compounds. Moreover, it is not clear if they act as photoprotective compounds, antenna pigments, or if they photoconvert part of the light spectrum to help zooxanthellae photosynthesis. These results are contested by some authors who excluded the role of phycoerythrin as a pigment compound in corals [5]. In order to determine the presence and the activity of cyanobacteria in corals, the following aspect should be considered: nonquantitative approaches cannot assure accurate values of abundance; moreover, the presence of *nifH* gene is not necessarily linked to the fixation and the transfer of nitrogen performed by diazotrophs. *H* [20]. Endolithic cyanobacteria have been found in *Porites cylindrica* and *Montipora monasteriata*, but their role in the relationship with host corals is unknown [162]. In contrast, in other cnidarians, it has been demonstrated that endolithic cyanobacteria establish symbiotic relationships with coral hosts: this is the case of *Plectonema terebrans*, a cyanobacterium belonging to the order Oscillatoriales [72]. Cold-water corals are ecosystem engineers providing a habitat for thousands of different species. Their trophism is related to the low energy, partially degraded, organic matter that derives from the photic zone of oceans [163]. To face the lack of nutrients, cold-water corals evolved, on one hand, from an opportunistic feeding strategy [164,165], and on the other hand, from a symbiosis with various diazotrophs, including cyanobacteria [166,167,168]. *Plectonema terebrans* filaments, visible as pinkish to violet staining, are able to colonize the entire skeleton of the cold-water corals *Desmophyllum dianthus* and *Caryophyllia huinayensis*; however, their density is higher at the skeleton portion covered with polyp tissue [72]. The close contact between coral tissues and cyanobacteria obliges the endoliths to exchange nutrients with the surrounding water through the polyp itself. This close relationship is advantageous for the cyanobacterium because the coral nematocysts protect it from the grazers [169], and it is mutualistic because such a close relationship inevitably includes exchanges of metabolites between organisms [170]. These metabolites produce benefits for the host and play a trophic and/or protective role in the symbiotic mutualistic relationship. Middelburg et al. suggested that in cold-water corals, a complete nitrogen cycle occurs similar to that inferred for tropical reefs, ranging from ammonium production and assimilation to nitrification, nitrogen fixation, and denitrification [166]. 

The effects of environmental changes on the nitrogen fixation rates are still poorly explored, especially if specifically related to the symbiotic diazotrophs and to cyanobacteria. Ocean acidification enhances nitrogen fixation in planktonic cyanobacteria, as in the case of *Crocosphaera watsoni*, due to enhancement of photosynthetic carbon fixation [171]. It is interesting to underline that in the planktonic diazotroph cyanobacterium *Trichodesmium* sp., which forms symbiotic association with diatoms [172], the nitrogen fixation is enhanced under elevated CO_2_ conditions [173], but it is strongly reduced if there is an iron limitation [174]. On the contrary, *Seriatopora hystrix* diazotrophs are sensible to ocean acidification, with a decline of the nitrogen fixation rate at high CO_2_ concentration, leading to consequences on coral calcification and potential starvation for both the coral and the *Symbiodinium* spp. [175]. In addition, environmental changes can increase in coral symbionts, the abundance of microbial genes involved in virulence, stress resistance, sulfur and nitrogen metabolisms, and production of secondary metabolites. These changes that affect the physiology of symbionts can also affect the composition of the coral-associated microbiota [74], with the substitution of a healthy-associated coral community (e.g., cyanobacteria, Proteobacteria), playing a key role in mediating holobiont health and survival upon disturbance [176], with a community related to coral diseases (e.g., Bacteriodetes, Fusobacteria, and Fungi). 

## 6. Ascidians and Other Tunicates

Tunicates are considered rich in biologically active secondary metabolites [177,178,179,180], but it is unclear if these bioactive compounds were produced by tunicates themselves or by associated microorganisms [181,182], although strong direct and indirect lines of evidence show that defensive compounds and other secondary metabolites are produced by various symbiotic prokaryotes and not by the tunicates themselves. Among tunicate symbionts, cyanobacteria have been found in symbiotic relationships with various tunicates, ranging from tropical to temperate environments. In fact, obligate associations with cyanobacteria of *Prochloron* and *Synechocystis* genus have been found in some species of ascidians belonging to the genera *Didemnum*, *Lissoclinum*, *Diplosoma*, and *Trididemnum* [77], with cyanobacterial cells distributed in the cavities and/or tunic [78]. These cyanobionts have been demonstrated to be part of the core microbiome, in which species and populations do not reserve the water–column ones and microbiome–host relationship is species specific and not correlated to the geographical location [9]. In colonial ascidians, such as *Botryllus schlosseri* and *Botrylloides leachii*, an abundant population of *Synechococcus*-related cyanobacteria have been identified [79], while in the Mediterranean ascidian *Didemnum fulgens*, a coral-associated cyanobacterium has been observed in its tissues [183]. In some cases, the cyanobiont completely or partially lacks the nitrogen-fixation pathway. This is the case of *Prochloron didemni*, in symbiosis with the tunicate *Lissoclinum patella*, which is probably involved in carbon fixation and in the ammonia incorporation and not in the nitrogen fixation [80,81]. In fact, in contrast with the presence of genes for the nitrate reduction pathway and all primary metabolic genes required for free-living, *Prochloron* seems to lack the capability to fix nitrogen and to live outside the host [80]. *Prochloron* sp. also protects the host versus active forms of oxygen, which can be formed during photosynthesis processes. The cyanobacterium produces a cyanide-sensitive superoxide dismutase, a Cu-Zn metalloprotein, that has been demonstrated to prevent the toxicity of superoxide radicals, hydrogen peroxide, and hydroxyl radicals in the host ascidians [82]. In *Lissoclinum patella*, other cyanobacteria were abundant in various tissues and one of these is *Acaryochloris marina*, a chlorophyll d-rich cyanobacterium, able to sustain oxygenic photosynthesis under near-infrared radiation that propagates through *Prochloron* cells and ascidian tissue [83]. The Caribbean tunicate *Trididemnum solidum* produces a peculiar biologically active molecule, the acyl-tunichlorine (Figure 2) [84,85], that contains both nickels accumulated by the tunicate and pheophytin, which is produced by organisms with photosynthetic machinery and suggests a dual origin of this compound. In fact, this tunicate hosts the cyanobacterium *Synechocystis trididemni*, which contributes to the production of acyl-tunichlorine synthesizing the pheophytin through an intermediate molecule, the pyropheophorbide [84,85]. In addition, behavioral tests demonstrated the presence of deterring compounds in ascidian larvae able to distaste predatory fishes. These compounds have been identified to be didemnin B (Figure 2) and nordidemnin [65]. Didemnin B was found in various tunicates, and it is similar to a bioactive molecule produced by other cyanobacteria, enforcing the idea that the predation-deterring compounds can be produced by cyanobionts [184], although the possibility of a horizontal gene transfer cannot be totally rejected [185,186]. The tunicate–cyanobacteria symbiosis is evidenced by the presence, in the host tunicate, of a cellulose synthase gene, similar to the one found in cyanobacteria, which probably derives from horizontal transfer between the two organisms [187,188] and that may have a role in the tunicates evolutive radiation and in the development of adult and larvae body plans [188,189,190]. The presence of a rich and bio-diversified microbiome makes tunicates promising models for various purposes and important for drug discovery [10,191].

## 7. Metabolic Interactions Involved in Symbiosis of Cyanobacteria

Greater insight into metabolic interactions between symbiont cyanobacteria and host organisms, particularly algae and sponges, could be useful for enhancing the growth efficiency of these organisms and their valuable bioactive compounds. Cyanobionts produce a large array of secondary metabolites, and symbiotic interactions could be a “unique ecological niche open space for evolution of novel metabolites” that are peculiar of the infochemical communication among these organisms [21]. In fact, some of these molecules are found only in prokaryotes in a symbiotic relationship with, for example, lichens, marine sponges, and beetle [27]. Environmental bioavailability of these bioactive secondary metabolites is lower than the ones used in these studies and, in addition, some of these molecules (e.g., nodularins) have been demonstrated to be produced intracellularly and liberated into the environment only during cell lysis. These lines of evidence suggest that it is unlikely these cyanobacterial bioactive molecules can play a role as allelopathic infochemicals and, consequently, their role in the symbiotic association is at least controversial. The possible role, suggested by some authors [21,192], could be linked to chemical defense against grazing, and it is demonstrated that at least some cyanobacterial molecules can enter the food webs and persist in the environment, having consequences on various target organisms. For example, the aforementioned nostopeptolide A (Figure 2) has been demonstrated to be a key regulator of hormogonia formation. The production and excretion of various nostopeptolide variants changed according to the symbiotic status, de facto regulating the *Nostoc* ability of infection and reconstitution of the symbiosis (Figure 4) [21,24]. Moreover, changes in the metabolomic profile, demonstrated, for example, in the case of *Nostoc-Gunnera* and *Nostoc-Blasia* interactions, have probably a key regulatory influence on hormogonia formation, affecting the infection. These chemoattractants, produced by host organisms, are hormogonia-inducing factors (HIFs), and their production seems to be stimulated by nitrogen starvation [193,194]. The production of HIFs is not peculiar of *Gunnera* and *Blasia*, and some of them have been identified in other species, for example, in the hornwort *Anthoceros punctatus* [195]. Investigations performed on different mutant strains of *Nostoc punctiforme* demonstrated that mutation of the *ntcA* gene reduced the frequency of HIF-induced hormogonia, leading to the incapacity to infect host organism [196]. On the contrary, strains that show a greater hormogonia induction in response to *Anthoceros* HIF also infect the plant at a higher initial rate than not-mutated strains. Various chemoattractants are produced by both host and nonhost organisms to attract hormogonia. In fact, these chemoattractants are sugar-based molecules, and it has been demonstrated that simple sugars, such as arabinose and glucose, are able to attract hormogonia [197]. In this context, the polysaccharide-rich mucilage secreted by mature stem glands of *Gunnera chilensis*, rich in simple sugar molecules and arabinogalactan proteins, could play a role in symbiosis communication with cyanobacteria, as demonstrated for other symbiotic relationships, i.e., *Alnus*–*Frankia* symbiosis [198]. Finally, in terrestrial species, it has been demonstrated that various lectins could act as chemoattractants, playing a crucial role in cyanobacterial symbiosis in bryophyte and *Azolla* species with cyanobacteria belonging to the *Anabaena* group [199], although they have probably been involved in fungus-partner recognition in lichens [199,200,201].

Other molecules are involved in symbiosis acting as hormogonia-repressing factors (HRFs). These repressing factors induce in *N. punctiforme* the expression of the *hrmA* gene that is part of the *hrmRIUA* operon. The *hrmRIUA* operon is similar to the uronate metabolism operon found in other bacteria, although hrma gene is peculiar of cyanobacteria with no sequence homology with any gene in the databases [194,202]. Other genes involved in the repression of the hormogonia formation are *hrmR*, which produce a transcriptional repressor, and *hrmE*, whose function is unknown and are negatively regulated by fructose [203]. Some authors conclude that fructose, or a converted form of this sugar that acts as an infochemical, might regulate hormogonia formation [204]. The synergistic interaction between host and cyanobacteria has been demonstrated in green algae coculture [205]. Although the cyanobacteria–green algae coculture influences growth, lipid, and nitrogen contents, it is interesting that various algae–cyanobacterium combinations led to the presence of peculiar secondary metabolites in the culture medium. According to the algae-cyanobacterium combination, from 6 to 45 new compounds are present in the culture medium, and many other secondary metabolites are absent if the individual cultures are compared.

The fact that the bouquet of volatile secondary metabolites secreted in the culture medium (secretome) of cocultures is peculiar of cyanobacterial strain indicates that this response of green algae is species-specific. This is confirmed by the observed phenomenon of growth-enhancing or inhibition on the components of the synergistic interaction, typical of each cocultured species. Volatile organic compounds, revealed by GC–MS analysis, such as hexanol, heptanone, tetradecane, pentadecane, heptadecane, etc., were present in all the investigated cocultivation and were also reported by other authors that investigated volatile organic compounds secreted in a symbiotic relationship, as in the case of the mentioned *Anabaena*-*Azolla* case [206]. Detected compounds have been demonstrated to have biological activities on the synergistic interaction and are part of the exchange of infochemicals that the two partners act to improve their physiological fitness, as in the case of hexadecane, which is involved in the regulation of central carbon metabolism and beta-oxidation of fatty acids [207], or trichloroacetic acid, which is involved in the incorporation of nitrogen in amino acids and proteins [208]. Lines of evidence suggested that signal–host interactions are related to the presence of various receptors belonging to the pattern recognition receptors (PRRs), and they include Toll-like receptors (TLRs), NOD-like receptors (NLRs), C-type lectin receptors (CTLRs) [209,210,211], G-protein coupled receptors (GPCRs), and peptidoglycan recognition proteins (PGRPs) [212,213]. PRRs recognize prokaryotic molecules such as cell surface molecules (i.e., lipopolysaccharide and peptidoglycan), while GPCRs and PGRPs recognize bacteria-derived molecules, such as signal peptides and short-chain fatty acids [212,213]. Although a few studies have been focused on the investigation of the relationship between cyanobacteria and host organisms, the presence of these receptors (except PGRPs) has been demonstrated in many invertebrates considered in this review, such as Porifera, Cnidaria, and Mollusca species [36]. In Porifera, the role of scavenger receptors cysteine rich (SRCRs) has been identified as regulators of host colonization by the microbiota. In fact, in *Petrosia ficiformis*, an SRCR gene acts as a mediator in the establishment of intracellular cyanobionts downregulated in sponge individuals living in dark caves in an aposymbiotic state and overexpressed in individuals living at a short distance in illuminated areas [214]. The same gene was identified in other symbiotic sponges, for example, in *Geodia cydonium*, and in species belonging to different phyla, such as the sea urchin *Strongylocentrotus purpuratus* [39].

## 8. Bioprospecting of Cyanobacteria Symbioses

Marine ecosystems, characterized by a vast range of environmental conditions and interactions among organisms, represent a huge repository of chemical diversity. Marine biotechnology aims at exploiting, in eco-sustainable ways, natural processes and biosynthetic pathways behind the chemical interactions among living marine species, for the identification of structurally diverse and biologically active secondary metabolites. In the last decades, more than 90 genera of cyanobacteria have been investigated for the biosynthesis of natural compounds belonging to several chemical classes, such as alkaloids, peptides, terpenes, polysaccharides, and polyketides. The cyanobacterial orders mainly studied are *Synechococcales*, *Nostocales*, *Chroococcales*, and *Oscillatoriales* [215]. The genus *Nostoc* synthesizes several variants of nostopeptolide, a cyclic heptapeptide, when cyanobacteria live in association with hosts. This group of compounds showed a strong antitoxin effect; nostopeptolides inhibited the transport of nodularin (70 nM) into hepatocytes (HEK 293); the blockage of nodularin uptake, through the organic anion-transporters OATP1B1/B3, avoided hepatotoxic-induced apoptosis [216]. Symbiosis can induce the production of cytotoxic molecules by cyanobacteria, such as nosperin (Figure 2) [27]. This compound is a chimeric polyketide and is a biosynthetic product of the trans-AT polyketide synthases [217]. This biosynthetic pathway has been elucidated firstly in heterotrophic bacteria associated with marine sponges, producing peridin-like compounds. These molecules demonstrated high toxicity for human cells; thus, they are considered interesting candidates for the development of new anticancer drugs [218,219]. Indeed, they can block proliferation in vitro of human promyelocytic cells (HL-60), human colorectal adenocarcinoma (HT-29), and human lung adenocarcinoma (A549) (mycalamides A and B (Figure 2) with IC_50_ < 5 nM). The mechanism of action of peridin-like compounds can be related to the interference of these compounds with protein biosynthesis and cell division processes [218]. 

Complete elucidation of chemical biosynthesis activated by the symbiotic relationship between cyanobacteria and other marine organisms can supply new information for new cocultivation approaches, improving the eco-sustainable production of molecules of interest. The food industry utilizes bacterial consortia to produce fermented food, improving food quality [220]. Cyanobacteria are known to exchange nutrients with host organisms (e.g., microalgae), and this can be used for the large-scale production of vitamins, such as vitamin B (Figure 2) [221]. The de novo synthesis of vitamin B_12_ is characteristic of certain prokaryotes. Cyanobacteria synthesize several vitamin B_12_ variants that, in a natural symbiotic relationship, are required by microalgae for their growth [222]. This cyanobacteria–microalgae relation can be optimized for the production of vitamins with applications in the nutraceutical industry. Another example of symbiotic interaction with biotechnological potential is the cyanobacteria–fungi association. Exopolysaccharides (EPSs) are produced by many fungal species and this group of compounds is responsible of immunomodulatory activity on the human immune system, via NF-кB and MAPK pathways [223]. The EPSs production can be implemented using the cocultivation of cyanobacteria with fungi. Angelis et al. [224] demonstrated that the production of EPS in coculture was higher (more than 30%) than the monocultures. Schmidt et al. identified patellamide peptides biosynthetic gene cluster in the obligate cyanobacterial symbiont *Prochloron didemni* [225] when in association with the ascidian *Lissoclinum patella* [225]. The in vitro effect of these cyclic peptides was already known since they induce cytotoxicity on human and murine cancer cells (murine leukemia cells, P388; human lung adenocarcinoma cells A549; human colorectal adenocarcinoma, HT-29) through inhibition (IC_50_ 2.5 pg mL^−1^) of topoisomerase II activity [226].

Cyanobacteria are considered potential cell farms for the natural production of pigment proteins, such as phycobilisomes (PBSs). PBSs act together to harvest light for photosynthetic apparatus; phycoerythrin (PE), phycocyanin (PC), allophycocyanin (APC), and phycoerythrocyanin (PEC) are the main proteins belonging to PBSs. These molecules were also found in cyanobacteria living in a symbiotic relationship with corals [4]. They mainly act as photoprotective compounds and exhibit in vitro beneficial effects, such as hepato-protective, antioxidant, anti-inflammatory, UV-screen, and anti-aging activities, making the cyanobacteria pigments an interesting class of compounds for their use in food, cosmetics, and pharmaceutical industries. Symbiosis can modify the biosynthetic rate of these pigments. Indeed, PE was found highly synthetized (> 71 gold particles μm^−2^, using the immunogold-labeling technique) [52], when dinoflagellate-cyanobacteria consortia were present in low nitrogen marine environments [109]. PE and PC were described as potent free radical scavengers [227,228]. In addition, PC exerted a strong antiproliferative effect on many human cancer cell lines. It triggered activation of Caspase 3 or 9 on HepG2 (human hepatoma, IC_50_ 100 μg mL^−1^ [229]), MCF-7 (breast cancer cells, IC_50_ 50 μg mL^−1^ [230]), Hela (cervical cancer cells, IC_50_ 80 μg mL^−1^ [231]), and SKOV-3 (ovarian cancer cell, IC_50_ 130 μM [232]). Same compound is also able to induce cell cycle arrest in cancer cells, such as HT-29 (colorectal adenocarcinoma, IC_50_ 30 μg mL^−1^ [233]), A549 (lung adenocarcinoma, IC_50_ 50 μg mL^−1^ [234]), K562 (erythroleukemic cells, IC_50_ 7 ng mL^−1^, [234], SKOV-3 (ovarian cancer cells, IC_50_ 160 μM [235]) and MDA-MB-231 (breast cancer cells, IC_50_ 10 μM [236]).

Cyanobacteria can contribute to sponge pigmentation and to the production of secondary metabolites, as defensive substances [134]. Several cyanobacterial strains were isolated from the Mediterranean sponge *P. ficiformis* [61]; some of these strains showed antiproliferative activity against human cells [61,135]. Aqueous extracts of isolated cyanobacteria (at 150 μg mL^−1^, final concentration) were used to treat two human cancer cell lines, Hela and SH-SY5Y (cervical cancer and neuroblastoma cell lines, respectively), detecting an antiproliferative effect soon after 6 h. The filamentous cyanobacterium *Oscillatoria spongeliae* produces a polybrominated biphenyl ether, when in association with the sponge *Dysidea herbacea*. The isolated compound 2-(2’, 4′-dibromophenyl)-4, 6-dibromophenol (Figure 2) revealed a strong antibacterial activity toward resistant bacterial pathogens (MIC ≤ 2.5 μg mL^−1^ [237]) and toxicity against other cyanobacteria, such as *Synechococcus* sp. strains. Another example of compound produced by cyanobacteria living in association with marine sponges is the cyclic heptapeptide leucamide A (Figure 2), isolated from the sponge *L. microraphis* [66]. This compound showed strong cytotoxicity against several tumor human cells [238]. In particular, the cyclic peptide was able to inhibit the proliferation of human gastric cancer cells (HM02), with a GI_50_ of 5.2 µg mL^−1^ and of two human hepatocellular carcinoma cell lines (HepG2, GI_50_ of 5.9 µg mL^−1^; Huh7, GI_50_ of 5.1 µg mL^−1^). These results are not surprising since several other cyclic peptides have been reported to be cytotoxic toward several similar cell lines [239]. William et al. isolated a cyclic depsipeptide named majusculamide C (Figure 2) from the sponge *Ptilocaulis trachys* [240]. This compound was found in cyanobacteria associated with the abovementioned sponge and revealed a strong antifungal activity against plant pathogens, such as *Phytophthora infestans* and *Plasmopora viticola* [66,241]. 

The cooperation between microorganisms and corals also produces chemical advantages for the host [154]. In particular, coral mucus is considered of great interest for its immunomodulatory properties [242]. Mucus chemical composition is influenced by photosynthetic symbionts, such as cyanobacteria. Coral mucus is rich in carbohydrates and contains glycoproteins, such as mucins, polysaccharides, and lipids [243]. Mucins showed no toxic effect on human cells (up to 500 µg mL^−1^) and exhibited potential immunomodulatory property. This glycoprotein family can activate antioxidant mechanisms and immune responses on RAW 264.7 macrophage cells and zebrafish embryos (concentration range 50–400 µg mL^−1^ [244]). UV rays represent one of the most harmful abiotic factors and organisms exposed to high levels of UV radiation often collaborate, through a symbiotic relationship, for the construction of a more efficacious defense mechanism. In this regard, cyanobacteria produce mycosporine-like amino acids (MAAs). They are UV-absorbing hydrophilic molecules that are considered promising for the formulation of skin care products [245]. MAAs can absorb light in the range of UV-A (315–400 nm) and UV-B (280–315 nm); this process does not produce dangerous compounds (e.g., free radicals). MAAs demonstrated strong in vitro scavenging activity (scavenging concentration SC_50_ of 22 μM) and exerted a protective effect on human cells (A375, concentration range 0.1–100 µM) against oxidative stress, induced by oxygen peroxide (H_2_O_2_, up to 25µM). The protective mechanism can be observed at the nucleus level, where MAAs, comparable to the well-known ascorbic acid, counteract the genotoxic effect of H_2_O_2_ (10 and 25 μM), which causes DNA strand breaks [246]. 

More than 300 new metabolites have been discovered in tunicates since 2015 [191,247]. Some cyanobacteria-associated bioactive compounds have been identified, such as patellamide A and C (Figure 2) [225,248,249,250], engineered and produced using *Escherichia coli*, and ulicyclamide and ulithiacyclamide (Figure 2), isolated in the 1980s in the tunicate *Lissoclinum patella* [251]. Ulicyclamide showed strong antiproliferative activity against leukemia cells (L1210, IC_50_ 7.2 μg mL^−1^). The same antiproliferative effect was found when human urinary bladder carcinoma cells (T24, IC_50_ 0.1 μg mL^−1^) and T lymphoblastoid cells (CEM, IC_50_ 0.01 μg mL^−1^) were treated with Ulicyclamide [252]. In addition, a wide variety of toxic cyclic peptides were isolated from *Prochloron* species, produced through a PRPS pathway [225,248,253] and some gene biosynthetic highly conserved clusters. The high variability of cyanobacterial bioactive compounds is caused by the hypervariability of precursor peptides cassettes [254]. In addition, *Prochloron* metagenomic analyses evidenced the presence of additional metabolite gene clusters that can be involved in the production of yet unknown bioactive compounds with defensive functions [255]. Another defense mechanism, typical of benthic marine organisms, is the production of deterring compounds against predators. Didemnin B (Figure 2), a cyclic depsipeptide, has been found in many tunicates; it inhibits the proliferation of MOLT-4 cells (human T lymphoblasts; IC_50_ 5 nM) through cell cycle arrest (G1/S phase) [256]. This compound did not reach the market for its cardiac and neuromuscular toxicities. However, the structurally similar molecule dehydrodidemnin B (aplidine, Figure 2), produced by the Mediterranean tunicate *Aplidium albicans*, exhibited more potent antiproliferative activity and less toxic nonspecific effects. This compound reached the phase II trials as anticancer drug against medullary thyroid carcinoma, renal-cell carcinoma, and melanoma [257,258]. The volatile organic compounds (VOCs) are bioactive metabolites produced by cyanobacteria and their in vitro biosynthesis is influenced by cocultivation conditions with symbiotic microorganisms. VOCs isolated from a strain of the genus *Synechococcus* showed antibacterial activity (50 mg mL^−1^ of the total extract) against the Gram-negative bacterium *Salmonella typhimurium* [259].

## 9. Conclusions

Although symbiosis was once discounted as an anecdotal evolutionary phenomenon, evidence is now overwhelming that obligate or facultative associations among microorganisms and between microorganisms and multicellular hosts had crucial consequences in many landmark events in evolution and in the generation of phenotypic diversity and complex phenotypes able to colonize new environments. The ability to reconstruct evolution at the molecular level, and especially comparative analyses of full genome sequences, revealed that integration of genes originating from disparate sources has occurred on a very large scale. Lateral gene transfer is clearly important in prokaryotes, but in many cases, and particularly in multicellular eukaryotes, the route to recruiting foreign genes, and thereby novel metabolic capabilities, involves symbiotic association, i.e., a persistent close interaction with another species. Symbiosis binds organisms from all domains of life and has produced extreme modifications in genomes and structure. Symbiosis affects genome evolution by facilitating gene transfer from one genome to another and the loss from one genome of genes present in both symbiotic partners. The result is a complex, fused (conceptually and often literally) meta-organism, with different compartments for different portions of its required genes, mechanisms for signaling between the partners and transporting gene products between compartments, and new combinations of metabolic pathways leading to biochemical innovation, as previously demonstrated. Parasitic interactions, which are considered symbiotic in that they involve intimate multigenerational association between organisms, are a conspicuous example of genomic interplay over evolutionary timescales and metabolic manipulation of one organism by other and have also led to the evolution of complex chemical defense mechanisms, including an extremely diverse panel of repellent or toxic secondary metabolites. For all these reasons symbioses, in particular, those involving cyanobacteria are thus a highly promising potential source of novel chemical entities relevant for the drug discovery process and the development of functional ingredients, with different fields of applications. 

Many studies reported in this review highlight how secondary metabolites produced by cyanobacteria can vary in terms of composition and abundance, depending on many abiotic and biotic factors; symbiotic relationship can strongly modify the activation of biosynthetic pathways, producing specific molecules. Elucidating environmental factors that govern growth, distribution, and interspecific interactions of cyanobacteria in marine environments could increase our knowledge and ability to induce the expression of bioactive molecules for drug discovery. A huge number of molecules, with promising biotechnological activities, has been reviewed in this work, from the symbiosis between cyanobacteria and a large plethora of marine organisms. They can find applications in the food, cosmeceutical, nutraceutical, and pharmaceutical industries. Here, we focused our attention on the symbioses of cyanobacteria with few phyla of organisms (fungi, bacteria, diatoms, macroalgae, seagrasses, sponges, tunicates) because these obtained sufficient attention in previous investigations. However, it is likely that focusing on the relationships of cyanobionts with other groups of invertebrates and microorganisms will provide evidence for novel cases of symbioses. Evidently, further research studies on the still poorly explored field of this particular kind of symbiosis will promote enriching the overabundance of active metabolites already reported. In addition, studies targeted at the development of novel genetic and metabolic tools aimed at their overproduction will strongly enrich the market with novel marine bioactive compounds.

## Figures and Tables

**Figure 1 marinedrugs-19-00227-f001:**
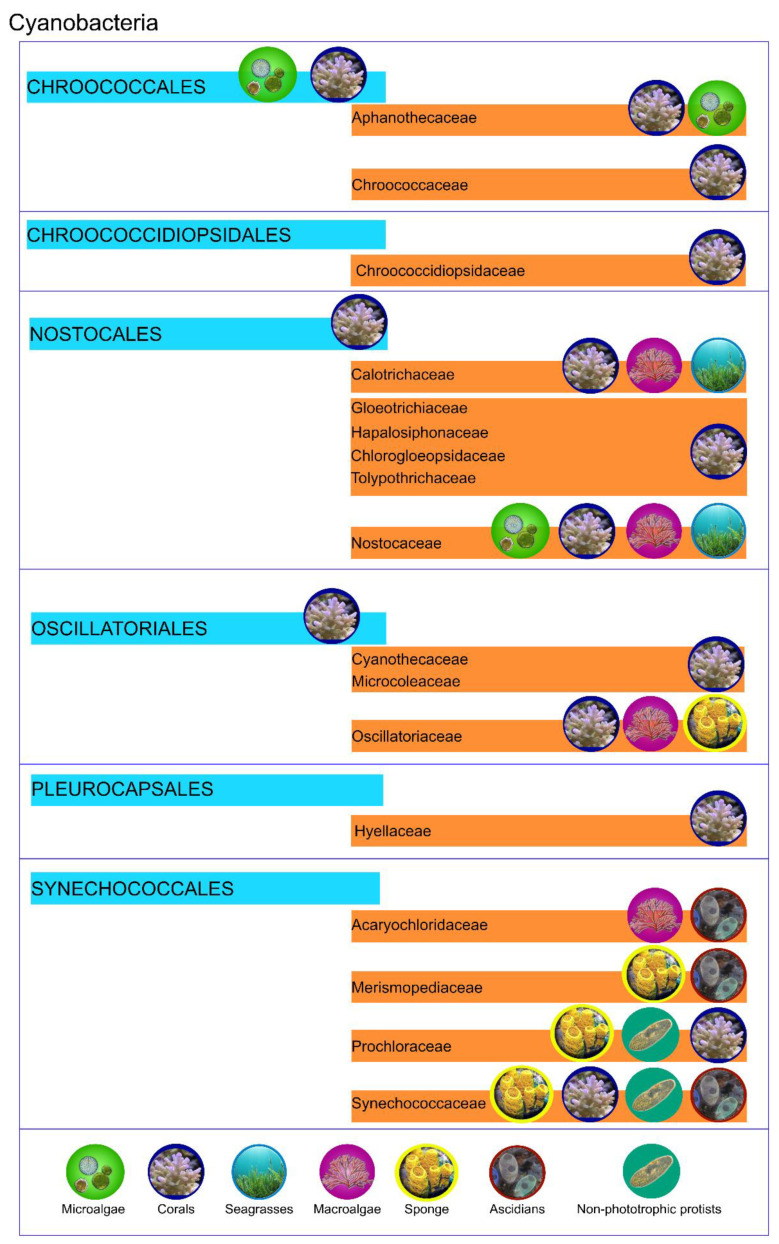
Symbioses of cyanobacteria. In this figure are summarized the symbioses among different cyanobacteria taxa with different hosts.

**Figure 2 marinedrugs-19-00227-f002:**
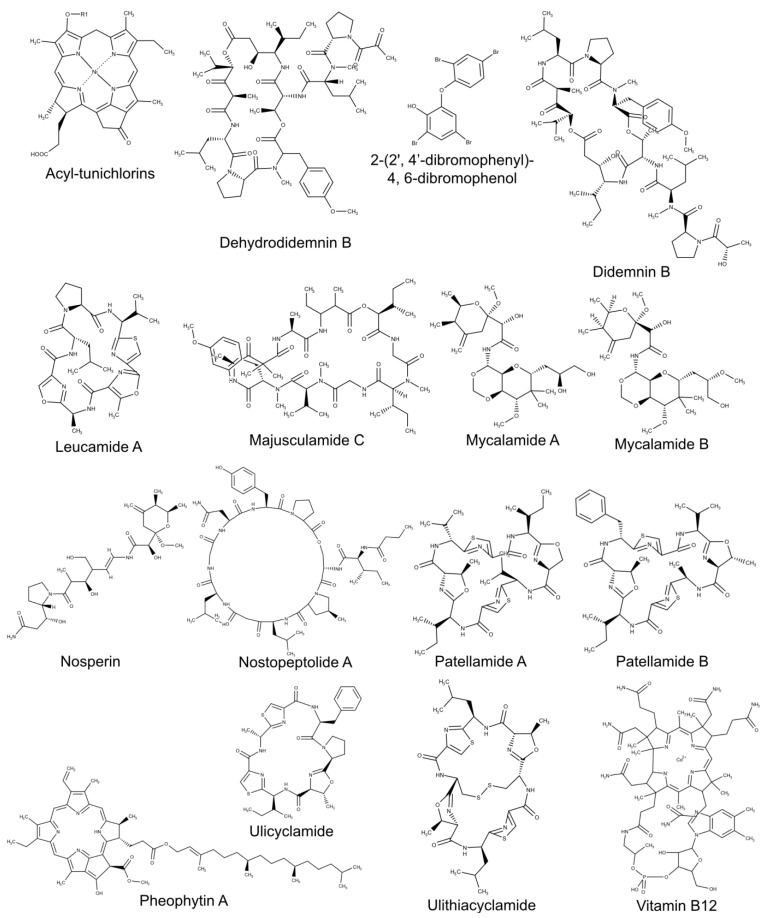
Structure of bioactive compound produced by symbiotic cyanobacteria.

**Figure 3 marinedrugs-19-00227-f003:**
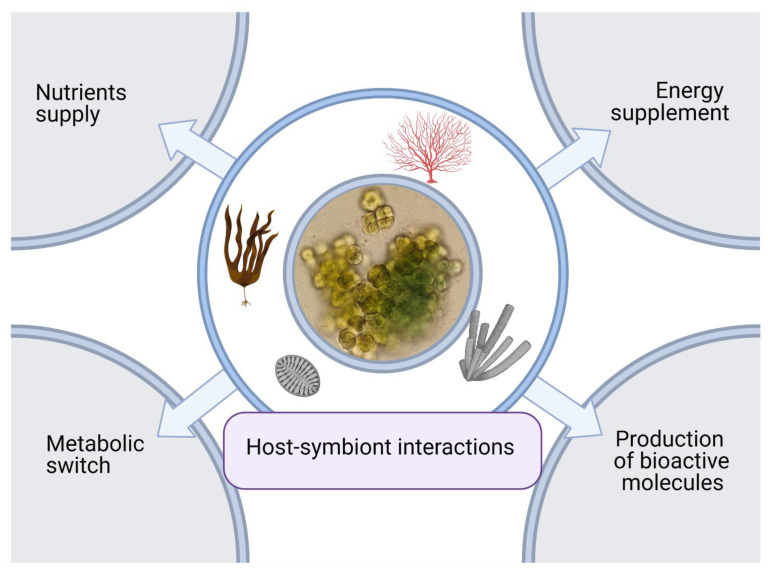
Ecological relevance of cyanobacteria in symbioses. Cyanobacteria symbioses have an important role in nutrient supply and energy supply, such as diazotrophy or photosynthesis. Cyanobacteria can also produce bioactive molecules that protect the host (i.e., anti-grazing compounds). In addition, the host can induce metabolic variation in cyanobacteria; indeed, several organisms are able to produce chemoattractants and hormogonia-inducing factors that allow symbiosis establishment and persistence.

**Figure 4 marinedrugs-19-00227-f004:**
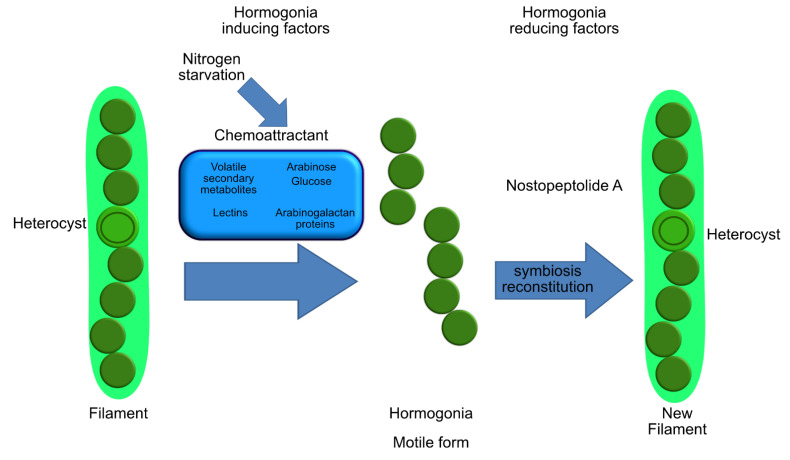
Schematic representation of hormogonia induction and repression in cyanobacterial symbiosis. Hormogonia motile forms, stimulated by several inducing factors that act as chemoattractants, are able to infect the host. Once infected, the host produces hormogonia-reducing factors, reconstituting the symbiosis.

**Table 1 marinedrugs-19-00227-t001:** Cyanobacteria and hosts involved in symbiotic interactions.

Host	Cyanobacteria	Interaction	Ref.
***Microalgae (or photosynthetic protists)***			
Bacillariophyta—*Rhizosolenia*, *Hemiaulus*, *Guinardia* and *Chaetoceros*	*Richelia intracellularis* and *Calothrix rhizosoleniae*	Nitrogen fixing	[18,40]
Bacillariophyta—*Climacodium frauenfeldianum*	*Crocosphaera watsonii*	Nitrogen fixing	[41]
Bacillariophyta—*Streptotheca* and *Neostrepthotheca*	*Crocosphaera watsonii*	Nitrogen fixing	[42]
*Solenicola setigera and* Bacillariophyta—*Leptocylindrus mediterraneus*	*Synechoccus* sp.	Nitrogen fixing and photosynthesis	[43,44]
Haptophyta—*Braarudosphaera bigelowii*	*Candidatus Atelocyanobacterium thalassa*	Nitrogen fixing. Cyanobacterium lack in oxygen-evolving photosystem II (PSII), RuBisCo for CO_2_ fixation, and tricarboxylic acid (TCA)	[45,46,47,48,49]
***Non-photosynthetic protists***			
Dinoflagellates	*Synechococcus* and *Prochlorococcus*	Nitrogen fixing	[50,51]
Tintinnids, Dinoflagellates, Radiolarians,	*Synechococcus*	Nitrogen fixing	[51,52]
***Macroalgae***			
*Ahnfeltiopsis flabelliformis*	*Acaryochloris marina*	Not reported	[53]
*Acanthophora spicifera*	*Lynbya* sp.	Nutrient supply	[54]
*Codium decorticatum*	*Calothrix*, *Anabaena* and *Phormidium*	Nitrogen fixing	[55,56]
***Seagrasses***			
*Thalassia testudinum*	unidentified	Carbon fixation	[57,58]
*Cymodocea rotundata*	*Calothrix*, *Anabaena*	Nitrogen fixing	[59]
***Sponge***			
*Petrosia ficiformis*	*Halomicronema metazoicum*	Not reported	[60]
*Petrosia ficiformis*	*Halomicronema* cf. *metazoicum*	Production of secondary metabolites	[61]
*Petrosia ficiformis*	*Cyanobium* sp.	Production of secondary metabolites	[61]
*Petrosia ficiformis*	*Synechococcus* sp.	Production of secondary metabolites	[61]
*Petrosia ficiformis*	*Pseudoanabaena* sp. *1*	Production of secondary metabolites	[61]
*Petrosia ficiformis*	*Pseudoanabaena* sp. *2*	Production of secondary metabolites	[61]
*Petrosia ficiformis*	*Leptolyngbya ectocarpi*	Production of secondary metabolites	[61]
*Petrosia ficiformis*	Undetermined Oscillatoriales	Production of secondary metabolites	[61]
*Petrosia ficiformis*	*Aphanocapsa feldmannii*	Food supply	[62,63]
*Chondrilla nucula*	Not classified	Feeding	[63]
*Dysidea herbacea*	*Oscillatoria spongeliae*	Defensive ecological role—production of toxic compounds	[64,65]
*Leucetta microraphis*	Not classified	Defensive ecological role—production of toxic compounds	[66]
*Ptilocaulis trachys*	Not classified	Defensive ecological role—production of toxic compounds	[66]
***Cnidaria***			
*Acropora hyacintus* and *A. cytherea*	*Synechococcus* and *Prochlorococcus*	Nitrogen fixing	[67]
*Montastraea cavernosa*	*Synechococcus* and *Prochlorococcus*	Nitrogen Fixing and Photoprotective or photosynthesis	[4]
*Acropora millepora*	Not classified	Nitrogen Fixing	[68,69,70]
*Porites astreoides*	Chroococcales, Nostocales, Oscillatoriales and Prochlorales	Nitrogen Fixing	[6]
*Acropora muricata*	Not classified	Not reported	[69]
*Pocillopora damicornis*	Not classified	Not reported	[69]
*Isopora palifera*	*Chroococcidiopsis* - Chroococcales	Nitrogen Fixing	[71]
*Montipora flabellate* and *M. capitate*	*Fischerella UTEX1931; Trichodesmium* sp.; *Lyngbya majuscule; Cyanothece* sp.*; Gloeothece* sp.*; Synechocystis* sp.*; Myxosarcina* sp.*; Leptolyngbya boryana; Chlorogloeopsis* sp*.; Calothrix* sp.; *Tolypothrix* sp.; *Nostoc* sp.; *Anabaena sphaerica.*	Nitrogen Fixing	[7]
*Desmophyllum dianthus*	*Plectonema terebrans*	Opportunistic feeding strategy	[72]
*Caryophyllia huinayensis*	*Plectonema terebrans*	Not reported	[72]
*M. cavernosa, M. franksi* and *Diploria and Porites* genus	*Anabaena, Synechococcus, Spirulina, Trichodesmium, Lyngbya, Phormidium* and *Chroococcales cyanobacterium*	Nitrogen Fixing Photoprotective compounds	[4,73,74,75,76]
***Ascidians***			
*Didemnum*, *Lissoclinum*, *Diplosoma* and *Trididemnum*	*Prochloron* and *Synechocystis*	Secondary metabolites production	[77,78]
*Botryllus schlosseri* and *Botrylloides leachii*	Synechococcus related	Secondary metabolites production	[79]
*Lissoclinum patella*	*Prochloron didemmi*	Carbon and ammonia fixing; Oxidative stress protection	[80,81,82]
*Lissoclinum patella*	*Acaryochloris marina*	Not reported	[83]
*Trididemnum solidum*	*Synechocystis trididemni*	Production of biologically active molecules	[84,85]

## Data Availability

The study did not report any data.

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
