# Peer review of "Symbioses of Cyanobacteria in Marine Environments: Ecological Insights and Biotechnological Perspectives"

_marinedrugs, 2021, doi:10.3390/md19040227_

Round 1

Reviewer 1 Report

Manuscript: The Cyanobacteria symbiosis in marine environments: ecological insights and biotechnological perspectives Generally, the manuscript is written well. It would be better if the authors could provide the figure to describe the ecological relevance of cyanobacteria symbiosis. Additionally, the following writing should be fixed as well. Line 510-512: “some of these molecules (i.e., microcystins) have been demonstrated to be produced intracellularly and liberated into the environment only during cell lysis. These evidences suggest that it is unlikely that microcystins and nodularins can play a role as allelopathic infochemicals and, consequently,..” Microcystin is produced by freshwater, not marine cyanobacteria. However, this work focuses on marine cyanobacteria. What is the aim of mentioning the microcystin produced by freshwater cyanobacteria. Line 748-750: “Elucidating environmental factors that govern growth, distribution and interspecific interactions of cyanobacteria could increment the success rate of drug discovery from marine environments.”. The word increment should be replaced with increase. But this sentence does not make sense, elucidating environmental factors has nothing significant to do with the drug discovery. Drug discovery requires numerous screening the bioactivity of metabolites. Regardless, this sentence should be corrected. Line 750-751: “A huge number of molecules, with promising biotechnological activities, has been reported in this work”. It should read that ….. has been reviewed in this work”, Line 754-756: “Here we focalised our attention on a few phyla of organisms (fungi, bacteria, diatoms, macroalgae, seagrasses, sponges, tunicates) because these ones obtained sufficient attention in previous investigations….” This read quite confusing and be fixed. This work actually focused on cyanobacteria, and why did the authors mentioned that they focus on other organisms such fungi, bacteria…..

Author Response

Referee 1:

Manuscript: The Cyanobacteria symbiosis in marine environments: ecological insights and biotechnological perspectives Generally, the manuscript is written well. It would be better if the authors could provide the figure to describe the ecological relevance of cyanobacteria symbiosis.

A new figure has been added to the manuscript (Figure 3), evidencing the ecological relevance of cyanobacteria in symbiosis.

Additionally, the following writing should be fixed as well. Line 510-512: “some of these molecules (i.e., microcystins) have been demonstrated to be produced intracellularly and liberated into the environment only during cell lysis. These evidences suggest that it is unlikely that microcystins and nodularins can play a role as allelopathic infochemicals and, consequently,..” Microcystin is produced by freshwater, not marine cyanobacteria. However, this work focuses on marine cyanobacteria. What is the aim of mentioning the microcystin produced by freshwater cyanobacteria.

Thanks for your observation. It is a very interesting point that, in my opinion, should be investigated in the future. In fact, although microcystins have not been identified as peptide in marine environments (if we want to exclude the presence of this toxins derived from river estuaries), it has been found that at least some marine cyanobacteria have the pathway for the microcystins production (Frazão, B., Martins, R., & Vasconcelos, V. (2010). Are known cyanotoxins involved in the toxicity of picoplanktonic and filamentous North Atlantic marine cyanobacteria?. Marine drugs8(6), 1908-1919.). In marine Leptolyngbya sp. and Oscillatoria sp., it was demonstrated the presence of 439 and 431 base pair (bp) fragments with 99% similarity with the mcyE of the microcystin synthetase gene cluster from a Microcystis sp. CYN06 strain. In any case, due to the controversial ability to produce such toxins, we will remove “microcystins” from the sentence, also because there are many other marine cyanotoxins with the same dynamic. Is this the case of nodularins, that are considered “mainly marine/brackish toxins produced by Nodularia spumigena” (Foss, A. J., Butt, J., Fuller, S., Cieslik, K., Aubel, M. T., & Wertz, T. (2017). Nodularin from benthic freshwater periphyton and implications for trophic transfer. Toxicon140, 45-59.)

Line 748-750: “Elucidating environmental factors that govern growth, distribution and interspecific interactions of cyanobacteria could increment the success rate of drug discovery from marine environments.”. The word increment should be replaced with increase. But this sentence does not make sense, elucidating environmental factors has nothing significant to do with the drug discovery. Drug discovery requires numerous screening the bioactivity of metabolites. Regardless, this sentence should be corrected.

Our sentence is quite misleading. It has been modified in “Elucidating environmental factors that govern growth, distribution and interspecific interactions of cyanobacteria in marine environments could increase our ability to identify potential candidate molecules for drug discovery.”. Thanks for your suggestion.

Line 750-751: “A huge number of molecules, with promising biotechnological activities, has been reported in this work”. It should read that ….. has been reviewed in this work”,

“Reported” has been substituted by “reviewed” as suggested.

Line 754-756: “Here we focalised our attention on a few phyla of organisms (fungi, bacteria, diatoms, macroalgae, seagrasses, sponges, tunicates) because these ones obtained sufficient attention in previous investigations….” This read quite confusing and be fixed. This work actually focused on cyanobacteria, and why did the authors mentioned that they focus on other organisms such fungi, bacteria…..

The sentence is, of course, not clear. We changed in “Here we focalised our attention on the symbioses of cyanobacteria with few phyla of organisms (fungi, bacteria, diatoms, macroalgae, seagrasses, sponges, tunicates) because these ones obtained sufficient attention in previous investigations. However, it is likely that focalizing on the relationships of cyanobionts with other groups of invertebrates and microorganisms will provide evidence for novel cases of symbioses.” Thanks for your suggestion.

Reviewer 2 Report

This manuscript provided an extensive review onThe Cyanobacteria symbiosis in marine environments:ecological insights and biotechnological perspectives . The information is new and interesting to the community. However, I do have a few questions and suggestions of revision before the manuscript get accepted.

line 35- 37 :Symbiotic relationships …or for both (mutualism) , please incorporate a separate reference for this information

 line 52-54 : In parallel, …. competition levels. Please rewrite this sentence

comment 1: “symbiotic associations with various taxa”,line 32 to 120 under this section most of the information provided is quite generalised

  • please try to make it crisp for the reader
  • try to reduce the generalised information.
  • Please Incorporate the more information justifying the title in the above said section

Comment 2:  figure 1 can me more illustrative, please try to upgrade the illustration

Comment 3 :This is a review paper with many references. The authors will need to better organize their language in some of the sections following a clear logic, rather than collecting and rephrasing the results of related studies.

Comment 4: it would be nice to see if the author’s incorporate a figure illustration here in this section  “Metabolic interactions involved in symbiosis of cyanobacteria.”

Comment 5: Some sections are coherently organized, but there is a lot of redundancy in the topics of discussion between different sections,  It is important that each section collate information from many sources to identify trends that are relevant to the topic of that section. 

Author Response

This manuscript provided an extensive review on The Cyanobacteria symbiosis in marine environments:ecological insights and biotechnological perspectives . The information is new and interesting to the community. However, I do have a few questions and suggestions of revision before the manuscript get accepted.

line 35- 37 :Symbiotic relationships …or for both (mutualism) , please incorporate a separate reference for this information

A Refence has been added to the sentence

 line 52-54 : In parallel, …. competition levels. Please rewrite this sentence

The sentence has been modified as “In addition, cyanobacteria in symbiosis have the advantages to be protected symbiosis protects cyanobionts from environmental extreme conditions as well as from predation/grazing and, at the same time,  hosting organisms grants enough space to cyano-bionts to grow at low competition levels.”

comment 1: “symbiotic associations with various taxa”, line 32 to 120 under this section most of the information provided is quite generalised

  • please try to make it crisp for the reader
  • try to reduce the generalised information.
  • Please Incorporate the more information justifying the title in the above said section

This first section was, from our point of view, a general introduction to the central parts where the topic is more thoroughly investigated. The title of the section is probably misleading and has been changed consequently. In addition, as suggested by the referee, the introductory section has been revised to make concepts more linear and easy to read.

Comment 2:  figure 1 can me more illustrative, please try to upgrade the illustration

We improved the figure n°1, making it more illustrative and clear. Some minor issues have been solved, taxa have been grouped together and a legend has been added.

Comment 3 :This is a review paper with many references. The authors will need to better organize their language in some of the sections following a clear logic, rather than collecting and rephrasing the results of related studies.

We improved the manuscript as suggested by the referee. Some sections have been deleted or moved in order to have a better organization along with the manuscript.

Comment 4: it would be nice to see if the author’s incorporate a figure illustration here in this section  “Metabolic interactions involved in symbiosis of cyanobacteria.”

Figure 3 has been added to the manuscript in the section “Metabolic interactions involved in symbioses of cyanobacteria.”

Comment 5: Some sections are coherently organized, but there is a lot of redundancy in the topics of discussion between different sections,  It is important that each section collate information from many sources to identify trends that are relevant to the topic of that section. 

We revised the manuscript, deleting redundancies and making it less fragmented and more consistent.